# *Dendrobium nobile* Lindley Administration Attenuates Atopic Dermatitis-like Lesions by Modulating Immune Cells

**DOI:** 10.3390/ijms23084470

**Published:** 2022-04-18

**Authors:** Sooyeon Hong, Eun-Young Kim, Seo-Eun Lim, Jae-Hyun Kim, Youngjoo Sohn, Hyuk-Sang Jung

**Affiliations:** Department of Anatomy, College of Korean Medicine, Kyung Hee University, 26, Kyungheedae-ro, Dongdaemun-gu, Seoul 02447, Korea; ghdtndus121@naver.com (S.H.); turns@daum.net (E.-Y.K.); hephaistion@naver.com (S.-E.L.); jhk1@khu.ac.kr (J.-H.K.)

**Keywords:** atopic dermatitis, IgE, IL-6, CD4, CD8, HaCaT, cytokine, TARC, GM-CSF, NF-κB

## Abstract

Atopic dermatitis (AD) is a chronic inflammatory skin disease that can significantly affect daily life by causing sleep disturbance due to extreme itching. In addition, if the symptoms of AD are severe, it can cause mental disorders such as ADHD and suicidal ideation. Corticosteroid preparations used for general treatment have good effects, but their use is limited due to side effects. Therefore, it is essential to minimize the side effects and study effective treatment methods. *Dendrobium nobile* Lindley (DNL) has been widely used for various diseases, but to the best of our knowledge, its effect on AD has not yet been proven. In this study, the inhibitory effect of DNL on AD was confirmed in a DNCB-induced Balb/c mouse. In addition, the inhibitory efficacy of inflammatory cytokines in TNF-α/IFN-γ-induced HaCaT cells and PMACI-induced HMC-1 cells was confirmed. The results demonstrated that DNL decreased IgE, IL-6, IL-4, scratching behavior, SCORAD index, infiltration of mast cells and eosinophils and decreased the thickness of the skin. Additionally, DNL inhibited the expression of cytokines and inhibited the MAPK and NF-κB signaling pathways. This suggests that DNL inhibits cytokine expression, protein signaling pathway, and immune cells, thereby improving AD symptoms in mice.

## 1. Introduction

Atopic dermatitis (AD) is a chronic inflammatory skin disease, characterized by erythema, psoriasis, lichenification, itching, and eczematous skin lesions [1]. AD can increase the likelihood of depression, anxiety, behavioral disorders, and attention-deficit/hyperactivity disorder (ADHD), and in severe cases, it can even cause suicidal thoughts [2]. Additionally, AD can cause fatigue due to reduced sleep time and increased sleep disturbance due to extreme itching, which affects daily life [3]. AD is caused by environmental factors, genetic factors, and an imbalance of immune cells. According to previous studies, AD has a prevalence of up to 22.6% in children and 17.1% in adults [4]. In addition, if the parent has a history of AD or high-risk occupations, the chance of developing AD in their children during adolescence is up to 81% [5]. AD affects families in a variety of ways, including long-term treatment, costs (more than $2000 per year), extreme stress, and, in severe cases, family disruption [6]. This significantly reduces the patient’s overall quality of life and can lead to social and national losses beyond the family.

Immunity consists of a balance of T helper (Th) 1 cells, which fight viruses and intracellular pathogens via a type-1 pathway called cellular immunity, and Th2 cells, which upregulate antibody production via a type-2 pathway called humoral immunity [7]. AD refers to damage to the skin barrier by causing Th1/Th2 immune imbalance through the excessive differentiation of Th2 cells and the expression of cytokines and chemokines, such as thymus, activation-regulated chemokine (TARC), and Th2-mediated interleukin (IL)-6 [8]. TARC is produced in various cells, such as endothelial cells, dendritic cells, and keratinocytes, and plays a role in inducing T cells to pass through endothelial cells and migrate to inflammatory lesions. Recently, the concentration of TARC was also measured to evaluate the severity of AD. IL-6 is secreted from various inflammatory cells, such as T lymphocytes and macrophages, and helps the maturation of Th2 cells [9]. IL-6 is expressed in large amounts from the lesions of atopic dermatitis and promotes inflammatory response by increasing the infiltration of inflammatory cells [10]. Therefore, measuring TARC and IL-6 levels is the main indicator that can confirm its potential as a therapeutic development study in AD. Currently, the most commonly used treatment for AD is the administration of topical corticosteroids, used by 60% of patients [11]. However, despite their beneficial effects, such as alleviating symptoms and reducing inflammation in AD, their long-term use leads to tolerance and side effects, such as epidermal atrophy, glaucoma, growth retardation, and impaired wound healing [12]. In order to reduce these problems, recent efforts have focused on the development of safe and effective therapeutic agents; in particular, interest in the natural physiological activity of plant extracts as an alternative therapeutic agent is increasing.

*Dendrobium nobile* Lindley (DNL) is a stem part of *Dendrobium nobile* Lindley (Orchidaceae). DNL, called Seokgok in Korea, has been widely used in traditional Korean medicine for its antipyretic, antiangiogenic, analgesic, and anti-aging effects, as well as for the treatment of diabetes and cardiovascular disease [13]. DNL contains ingredients that show various pharmacological effects, among them gigantol, dendrobin, moupinamide, isoliquiritin, etc. [14]. Gigantol was known to suppress the expression of IL-6 and immunomodulate the signaling pathways of mitogen-activated protein kinase (MAPK) and nuclear factor kappa-light-chain-enhancer of activated B cells (NF-κB) [15,16]. Dendrobin has been reported to inhibit the mechanisms of inflammatory cytokines and NF-κB in inflammatory bowel disease (IBD) [17]. In addition, moupinamide inhibited inflammatory cytokines such as IL-6 and IL-1 beta (β) and inhibited NF-κB transcription into the intracellular nucleus [18]. Isoliquiritin depresses inflammatory response and promotes angiogenesis, thereby promoting skin wound healing [19]. Since the anti-inflammatory effect and the improvement of atopic symptoms are closely related [20], the anti-inflammatory effect of ingredients predicted that DNL had the potential to improve AD.

Therefore, this study aims to prove the efficacy of DNL in AD based on the anti-inflammatory effects of active ingredients in DNL. In this study, we investigated the effect of DNL in a chloro-2,4-dinitrobenzene (DNCB)-induced Balb/c mouse model, TNF-α/IFN-γ-induced HaCaT cells, and PMA plus A23187 (PMACI)-induced HMC-1 cells. Furthermore, we confirmed that DNL offers potential as an AD therapeutic agent.

## 2. Results

### 2.1. Effects of DNL on AD-like Symptoms in Balb/c Mice

The schedule of animal experiments to construct a DNCB-induced AD mouse model is shown in Figure 1. In this study, body weight, liver weight, and serum aspartate aminotransferase (AST) and alanine aminotransferase (ALT) data were measured to prove that there was no toxicity at low or high concentrations of DNL. During the animal experiment period, there was no significant increase in body weight in the Balb/c mice in the normal, control, DNL_L (1 mg/mL), and DNL_H (10 mg/mL) groups (Figure 1B). The toxicity of DNL was evaluated in animal experiments. On the day of sacrifice, the livers were collected and weighed. The application of DNL did not affect the liver weight (Figure 1C). When the levels of ALT and AST in serum were measured, both indicators were significantly decreased in the DNL_L group compared to the control group (Figure 1D,E). As a result of measuring the scratching and scoring on the Atopic Dermatitis (SCORAD) index after the repeated application of DNCB, the scratching behavior was found to have significantly increased in the control group compared to the normal group and to have decreased significantly in the DNL_L group compared to the control group (Figure 1F). The SCORAD score was significantly decreased in both the DNL_L and DNL_H groups compared to the control with DNCB (Figure 1G,H).

### 2.2. Effects of DNL on Immunoglobulin E (IgE) and IL-6 Levels and Mitogen-Activated Protein Kinase (MAPK) Protein Expression in Serum and Dorsal Skin

We performed ELISA to confirm the expression of IgE in DNCB-induced Balb/c mouse serum. The level of IgE was significantly increased in the control group compared to the normal group. However, in the DNL_H group, the IgE level significantly decreased compared with the control group (Figure 2A). We performed ELISA to evaluate the expression levels of IL-6 and IL-4 in dorsal skin tissue. The IL-6 expression was significantly increased in the control group compared to the normal group. However, the expression of IL-6 was significantly decreased in DNL_L and DNL_H compared to the control group (Figure 2B). The level of IL-4 was significantly increased in the control group compared with the normal group. DNL_L and DNL_H decreased compared to the control but not significantly (Figure 2C). The protein expression of MAPK was verified by Western blotting (Figure 2D). The protein expression of ERK was significantly increased in the control group compared to the normal group and decreased significantly in the group treated with DNL_H compared to the control group. The expression of p38 protein was significantly increased in the control group compared to the normal group in the same manner as in the ERK and decreased significantly in a DNL concentration-dependent manner compared to the control group (Figure 2E).

### 2.3. Effects of DNL on the Mast Cell, Eosinophil, Skin Thickness, CD4, and CD8 in DNCB-Induced AD-like Skin

We performed histopathological observations to measure the thickness of the skin and the infiltration of inflammatory cells. Figure 3A shows the observed thickness of the epidermis and dermis through H&E staining. As a result of the experiment, epidermis thickness was significantly increased in the control group compared to the normal group. However, when DNL_L and DNL_H were applied, the epidermal thickness significantly decreased compared to the control group (Figure 3D). The dermis thickness was significantly increased in the control group compared to the normal group in the same way as in the epidermis. Furthermore, the dermis thickness was significantly decreased in the DNL_L and DNL_H group compared to the control group (Figure 3E). Toluidine blue staining was performed to confirm mast cell infiltration (Figure 3B). The mast cells were counted after three-field counting at 200× magnification per tissue. In the control group, the infiltration of mast cells was significantly increased compared with the normal group. Furthermore, the DNL_L and DNL_H groups were significantly inhibited compared to the control group (Figure 3F). H&E staining was performed by checking the degree of infiltration of immune cells, eosinophils (Figure 3C). The eosinophil infiltration was significantly increased in the control group compared to the normal group. The DNL_L and DNL_H groups featured significantly reduced infiltration of eosinophils in the skin tissue compared with the control group (Figure 3G).

We confirmed the expression of CD4 and CD8 in DNCB-induced Balb/c mice by performing immunohistochemical (IHC) staining (Figure 4A). The CD4 infiltration was significantly increased in the control group compared to the normal group. However, when DNL_L and DNL_H were applied, there was a concentration-dependent decrease compared to the control group. In particular, DNL_H significantly reduced CD4 infiltration (Figure 4B). The infiltration of CD8 was confirmed; similarly, to the experimental results for CD4, the control group significantly increased compared to the normal group. However, when DNL_L and DNL_H were applied, the expression of CD8 was decreased in a concentration-dependent manner compared to the control group. In particular, the infiltration of CD8 was significantly reduced in the DNL_H group (Figure 4C).

### 2.4. Effects of DNL on Cytokines and Chemokines in HaCaT Cells

A cell counting kit (CCK)-8 was used to evaluate the toxicity of DNL in HaCaT cells. As a result of the experiment, 500 μg/mL of DNL had an effect on the cell viability in HaCaT cells, and the concentrations of DNL used in this study (62.5, 125, 250 μg/mL) did not affect the viability of the HaCaT cells (Figure 5A). In order to confirm the expression of inflammatory cytokines in HaCaT cells, mRNA was isolated from HaCaT cells. Reverse transcription quantitative polymerase chain reaction (RT-PCR) was performed on all the cDNA samples using TARC and IL-6 primers (Figure 5B). TARC and IL-6 significantly increased in the HaCaT cells stimulated with TNF-α/IFN-γ. TARC expression was significantly decreased in 125 and 250 μg/mL DNL (Figure 5C). IL-6 expression was significantly decreased in 250 μg/mL DNL (Figure 5D).

### 2.5. Effects of DNL on Inflammatory Cytokine and Chemokine in HaCaT Cell Culture Medium

In previous studies on AD, dexamethasone (Dex) and silymarin (Sily) were used as positive controls [21,22]. Therefore, in this study, the pharmacological effects of DNL were comparatively analyzed using Dex and Sily. ELISA was performed to investigate the effect of DNL on inflammatory cytokines and chemokines in HaCaT cell culture medium. DNL 125 and 250 μg/mL significantly reduced the expression of TARC, and the inhibitory ability of DNL was similar to that of the Sily. However, Dex significantly increased the level of TARC (Figure 5E). The level of IL-6 expressed was significantly increased in the HaCaT cells stimulated with TNF-α/IFN-γ. However, DNL 125 and 250 μg/mL significantly reduced the expression of IL-6. These results were similar to those of the Dex. The Sily did not significantly inhibit the expression of IL-6 (Figure 5F). The HaCaT cells stimulated with TNF-α/IFN-γ had increased granulocyte-macrophage colony-stimulating factor (GM-CSF) expression. However, 62.5, 125, and 250 μg/mL of DNL significantly reduced the expression of GM-CSF, which showed a similar inhibitory ability to the Dex (Figure 5G). The level of monocyte chemoattractant protein (MCP)-1 expression was significantly increased in HaCaT cells stimulated with TNF-α/IFN-γ. However, DNL 62.5, 125 and 250 μg/mL significantly reduced MCP-1 expression. Dex and Sily significantly inhibited the expression of MCP-1 (Figure 5H). The HaCaT cells stimulated with TNF-α/IFN-γ had significantly increased levels of TNF-α, whereas the expression of the TNF-α level in 125, 250 μg/mL DNL-treated HaCaT cells was significantly reduced. This was similar to the inhibitory ability of the Dex. However, the Sily did not suppress the expression of TNF-α (Figure 5I).

### 2.6. Effect of DNL on the Expression of Inflammatory Cytokine and Chemokine in HMC-1 Cell

A 3-(4,5-dimethylthiazol-2-yl)-5-(3-carboxymethoxyphenyl)-2-(4-sulfophenyl)-2h-tetrazolium, inner salt (MTS) assay was performed to evaluate the degree of toxicity of DNL in HMC-1 cells. In HMC-1 cells, 500 μg/mL of DNL showed cytotoxicity, but the DNL of 200 μg/mL or less used in our study showed no cytotoxicity (Figure 6A). We confirmed the expression of inflammatory cytokine and chemokine by ELISA in HMC-1 cells. Expression of IL-6 was significantly increased in PMACI-induced HMC-1 cells. DNL 200 μg/mL significantly reduced the expression of IL-6. However, the Dex and Sily, did not significantly inhibit the expression of IL-6 (Figure 6B). The GM-CSF expression was significantly increased in the HMC-1 cells stimulated with PMACI. However, the DNL 100 and 200 μg/mL has significantly inhibited the expression of GM-CSF, which showed a similar inhibitory ability to the Sily (Figure 6C). The level of MCP-1 expression was significantly increased in PMACI-induced HMC-1 cells. MCP-1 expression was significantly inhibited at 200 μg/mL of DNL. However, Dex and Sily could not inhibit the expression of MCP-1. These results indicate that DNL has superior efficacy in inhibiting the expression of MCP-1 than the positive control group (Figure 6D). The expression of TNF-α was significantly increased in PMACI induced HMC-1 cells. However, 100 and 200 μg/mL of DNL, Dex, and Sily significantly inhibited the expression of TNF-α (Figure 6E).

### 2.7. Effect of DNL on TNF-α/IFN-γ-Treated HaCaT Cell Activation of NF-κB and MAPKs

We investigated the effect of DNL on the protein expression of MAPK in HaCaT cells stimulated with TNF-α/IFN-γ (Figure 7A). Extracellular-signal-regulated kinase (ERK) protein expression was significantly inhibited by DNL at 125 μg/mL in the HaCaT cells stimulated with TNF-α/IFN-γ. In addition, c-jun n-terminal kinase (JNK) significantly inhibited the HaCaT cells stimulated with TNF-α/IFN-γ in DNL 62.5 μg/mL. However, the protein expression of p38 was not significantly inhibited by DNL (Figure 7B). After checking the markers of MAPK, it was confirmed whether DNL inhibits the NF-κB/IκB signaling pathway downstream of the MAPK signaling pathway (Figure 7C). The phosphorylation of p-NF-κB was confirmed in nuclear and was significantly increased in the HaCaT cells stimulated with TNF-α/IFN-γ. When DNL was applied, the phosphorylation of p-NF-κB was significantly reduced at 62.5, 125, and 250 μg/mL. In addition, the IκB in the cytosol significantly decreased the HaCaT cells stimulated with TNF-α/IFN-γ. The DNL was significantly increased at 62.5, 125, and 250 μg/mL concentrations (Figure 7D).

### 2.8. Liquid Chromatography–Mass Spectrometry (LC–MS) Analysis of DNL

In this study, the quality and purity of DNL were measured by LC-MS analysis. The chromatography profile of gigantol standard and DNL was shown in Figure 8A,B. The retention time of gigantol in DNL was 42–43 min. As a result of the analysis, the DNL has an 11.589 ppm content of gigantol.

## 3. Discussion

In this study, DNL showed anti-inflammatory effects on the pathological and histological symptoms of AD-like lesions in DNCB-induced Balb/c mice. In addition, DNL significantly inhibited inflammatory cytokines, MAPK, and NF-κB/IκB signaling pathways in TNF-α/IFN-γ-induced HaCaT cells and PMACI-induced HMC-1 cells.

The mouse is the most commonly used vertebrate animal due to its availability, size, low cost, ease of handling, and rapid reproduction rate. In addition, since the mouse shares 99% of its genes with humans, it has been widely used as a major model for human diseases [23]. In particular, the Balb/c mouse is widely used in the fields of immunology and pharmacology and has a high Th2 immune response, thereby inducing AD [24]. There are two main types of in vivo model study for atopic dermatitis studies: genetic mutants; and disease induction models using allergens, microbial antigens, and chemical reagents. Of the two, the model using chemical reagents was proven to be excellent in both reproducibility and economy [25]. The repeated application of DNCB to BALB/c mice shows characteristic symptoms of AD, such as high serum IgE levels, epidermal proliferation, and the induction of mast cell infiltration [26]. The DNCB-induced Balb/c model is a commonly used animal model to study the pathogenesis of dermatitis. Approximately 24 h after exposure to DNCB, macrophages were found to accumulate at the site of exposure to DNCB. Furthermore, the induction of DNCB initiates the pathogenesis of dermatitis, mainly due to T cell-mediated immune response [27]. Therefore, we confirmed the efficacy of DNL in vivo through the DNCB-induced Balb/c mice model.

AD has various symptoms, the most important of which is severe itching. Excessive itching and repeated scratching break down the skin barrier and increase the inflammatory response. The SCORAD index is a clinical score widely used to standardize the signs and symptoms of AD by scoring each of five items, such as dry skin, edema, eczema, and lichen [28]. By scoring AD patients with the SCORAD index, the severity of the disease can be checked and appropriate treatment can be suggested. Therefore, it is important to measure the SCORAD index in atopic dermatitis [29]. In this study, scratching behavior and the SCORAD index were significantly reduced when DNL was applied to DNCB-induced Balb/c mice. These results suggest that DNL improves the symptoms of AD.

According to a previous study, it is known that scratching behavior and SCORAD levels in clinical practice are closely related to the expression of IgE and IL-6 [30]. IgE induces the degradation of mast cells infiltrating the skin as the most representative inflammatory marker, whose level increases in the serum of patients with AD [31]. IL-6 increases the production of IgE, which induces mast cell degranulation, and induces Th2 maturation to induce inflammatory responses, such as eosinophil infiltration and epidermal hyperkeratosis [32]. The IL-4 is a cytokine produced by Th2 cells, natural killer (NK) T cells, eosinophils, mast cells, and activated basophils, which are central to pathogenesis and major therapeutic targets of AD [30]. IL-4 induces the production of IgE in B cells and plays a role in differentiating naive CD4^+^ T cells into Th2 cells [30]. In this study, we observed that DNL application significantly decreased the expression levels of IgE and IL-6 in DNCB-induced Balb/c mice. IL-4 expression was decreased in DNCB-induced Balb/c mice, but the results were not significant. Our results may suggest the possibility that the application of DNL suppresses the expression of inflammatory cytokines, thereby reducing the degree of scratching behavior and the SCORAD index.

The skin is the most external organ in the human body and provides a physical and functional barrier that prevents allergens and microorganisms from penetrating into the human body. Defects in the skin barrier are the most important pathological findings in atopic dermatitis. The epidermis and dermis, which are components of the skin, not only increase the infiltration of immune cells, such as mast cells and eosinophils, in AD-like lesions due to continuous exposure to DNCB, but also increase tissue thickening and epidermis and dermis thickness [33]. In addition, when T-cell-mediated inflammatory cells penetrate into the skin tissue, edema and thickening occur, increasing the thickness of the epidermis and dermis [34]. Mast cells are most commonly found in tissues that are easily exposed to external environments, such as the skin and airways, but are found in almost all tissues. Mast cells are degranulated by the binding of high-affinity IgE receptor (FcεRI) and IgE on the surface [35]. When degranulation occurs, histamine and inflammatory cytokines, such as IL-6, are released, causing severe itching, redness of the skin, vasodilation, and increased vascular permeability, which is known to play a central role in allergic reactions [36]. Eosinophils are the most commonly known cells in the immune system, and they act as mediators of parasite defense and allergies, such as basophils. Eosinophils are increased in atopic dermatitis lesions to secrete various inflammatory cytokines and chemokines. These secreted inflammatory cytokines and chemokines recruit pro-inflammatory cells to the skin lesion, further exacerbating the symptoms [37]. DNL inhibited mast cell and eosinophil infiltration by DNCB-induced BALB/c mice; in addition, DNL inhibited DNCB-induced epidermis and dermis thickness. The results suggest that DNL reduces skin thickness through the inhibition of immune cell infiltration in DNCB-mediated AD-like lesions.

We confirmed the activity of T cells, which play the most important role in the pathogenesis of atopic dermatitis. T cells protect the skin from pathogens, and the cytokines generated from these cells have antibacterial activity, which can reduce bacterial infection and pathogen colonization in patients with atopic dermatitis [38]. In the peripheral immune system, T lymphocytes are defined as CD4^+^ T cells and CD8^+^ T cells by the expression of the glycoproteins CD4 and CD8 on the cell surface. CD4 is expressed on Th2 helper cells and CD4 regulatory T cells, and CD4 binds to histocompatibility complex class (MHC) 2 on antigen-presenting cells (APCs) [39]. CD8 is expressed on cytotoxic T lymphocytes to bind histocompatibility complex class (MHC) 1 molecule [40]. The histocompatibility complex class (MHC) associated with CD4 and CD8 enhances T cell signaling and, in turn, increases inflammatory cytokine production, thereby enhancing the inflammatory response [41]. As a result of this study, DNL significantly inhibited the infiltration of CD4^+^ T cells and CD8^+^ T cells in DNCB-induced Balb/c mouse dorsal skin. Therefore, these results suggest that DNL can inhibit the activation of T cells and suppress the inflammatory response that occurs in AD.

Keratinocytes make up the majority of epidermal cells. Keratinocytes mostly perform the structural barrier function of the skin, but also play a role in the initiation of inflammation and immune responses in the skin and wound healing. However, keratinocytes have not been easily studied in vitro due to various factors such as the need for additional growth factors [42]. Therefore, we used HaCaT cells, which are naturally immortalized human keratinocytes that do not require these factors. HaCaT cells do not require additional growth factors for proliferation, and continuous differentiation is possible. HaCaT cells also exhibit major surface markers and functional activity of keratinocytes [43]. TNF-α and IFN-γ are pro-inflammatory cytokines, and when treated with HaCaT cells, they induce the secretion of inflammatory cytokines, thereby amplifying or promoting the inflammatory response [44]. HMC-1 cells are established from mast cell leukemia patients, and these cells exhibit characteristics of tissue mast cells, such as the expression of histamine, tryptase, heparin, and cell surface-antigen-profile [45]. In addition, HMC-1 cells release various allergens, cytokines, and chemokines through degranulation due to the stimulation of PMACI [46]. On the basis of these properties, HMC-1 cells have been widely used in the study of human mast cells. Therefore, in this study, the efficacy of DNL was confirmed in vitro using HaCaT cells stimulated with TNF-α/IFN-γ and HMC-1 cells stimulated with PMACI models.

Chemokines are expressed in keratinocytes and are known to control the migration of antigen-presenting cells and the recruitment and activation of leukocytes [47]. These chemokines include TARC, GM-CSF, and MCP-1. The TARC, Th2-related chemokine, is mainly found in the skin and plasma of patients with atopic dermatitis and is an important indicator of the development of inflammation. TARC is secreted by the activation of dendritic cells and dermal cells and induces immune cells to penetrate into inflammatory lesions [48]. These stimuli also induce other inflammatory cytokines, such as IL-6. The GM-CSF induces the recruitment and generation of granulocytes and macrophages to the site of inflammation in the early stages of inflammation [49]. MCP-1 is a member of the CC chemokine family and is produced by epithelial, endothelial, and fibroblast cells, and recruits monocytes, memory T cells, and dendritic cells to inflammatory lesions [50]. TNF-α is a systemic inflammatory cytokine and has the ability to kill tumor cells in atopic dermatitis in vitro. In addition, the expression of TNF-α protein regulates the expression of MAPK and regulates psoriatic skin lesions [51]. As a result, DNL significantly inhibited TARC, IL-6, MCP-1, GM-CSF, and TNF-α in TNF-α/IFN-γ-induced HaCaT cells. In addition, the expression of GM-CSF, TNF-α, MCP-1, and IL-6 was significantly reduced in PMACI-induced HMC-1 cells. In particular, IL-6 showed the same pattern as the results obtained through the ELISA experiment in the serum of Balb/c mice. The results indicated that DNL significantly inhibited a representative index analyzing the therapeutic effect of AD. In addition, it can be suggested that DNL’s inhibition of intracellular inflammatory cytokine expression occurs through the inhibition of immune cells such as mast cells, eosinophils, CD4, and CD8 in animal experiments.

Positive controls are often used to judge test effectiveness of medicine. For example, to determine the suitability of a new alternative medicine to treat a disease, it can be compared to a commonly used medicine for that disease. Thus, we compared the pharmacological effects of DNL using Dex and Sily as positive controls. In HaCaT cells, Dex suppressed expression of IL-6, GM-CSF, MCP-1, and TNF-α, but not TARC. In addition, Sily inhibited expression of TARC, MCP-1, IL-6, and GM-CSF, but not TNF-α. On the other hand, DNL showed the effect of inhibiting all these cytokines and chemokines. In addition, in the HMC-1 cell experiment, DNL significantly inhibited GM-CSF, MCP-1, and TNF-α expression, but Dex only suppressed the expression of TNF-α. Additionally, Sily inhibited only GM-CSF and TNF-α. These results mean that DNL has the potential to be a better therapeutic alternative compared to Dex and Sily, which are used for the existing AD treatment.

In the MAPK signaling pathway, MAPK consists of ERK, JNK, and p38 [52]. MAPK is phosphorylated through a chain reaction in the order of MAPK kinase kinase (MAPKKK), MAPK kinase (MAPKK), and MAPK, and plays a role in triggering a cellular response by receiving external stimuli and transducing intracellular signals [53]. MAPK’s activation responds to various factors, such as pro-inflammatory cytokines, mitogens, heat shock, cell growth factors, and stress stimuli, and shows a high expression level in AD [54]. In this study, when DNL was applied to the skin of Balb/c mice, it inhibited the phosphorylation of ERK and p38, thereby inhibiting the MAPK signaling pathway. In HaCaT cells, the phosphorylation of ERK and JNK was significantly inhibited when treated with low DNL concentrations, but was not significantly inhibited at high concentration.

The insufficient effect of MAPK at the cellular level compared to the results of MAPK in animal experiments is seen as a difference in the trigger mechanism, depending on the stimulant. In the case of the DNCB used in animal experiments, it was found in previous studies that it elevates various factors, such as inflammatory cytokines, cell growth factors, and the stress applied to cells, thereby causing an inflammatory response [55]. On the other hand, in cell experiments, most of published studies were performed on the increase in the target for pro-inflammatory cytokines. Therefore, it is supposed that DNL inhibited other factors than the effect on inflammatory cytokines. However, in order to prove this theory, it seems that it would be meaningful to check the expression of indicators such as RAS, tumor necrosis factor receptor–associated factor (TRAF), RAC family small GTPase 1 (RAC 1), MAPK/extracellular protein kinase kinases (MEK), and MAPK kinase (MKK) in tissues.

NF-κB is a transcription factor that regulates the expression of inflammatory genes and is expressed by various factors. The nuclear translocation of NF-κB is initiated when the IκB protein is phosphorylated by cellular stimulation. NF-κB bound to IκB migrates to the nucleus, binds to target genes, and induces inflammatory cytokines, such as TARC, IL-6, and GM-CSF.

The expression of NF-κB is activated by the MAPK signaling pathway but is also stimulated by two other mechanisms. The two mechanisms are as follows: (1) TNF-α binds to the tumor necrosis factor receptor (TNFR), the TNFR1-associated death domain protein (TRADD) is mobilized, and the NF-κB kinase inhibitor (IKK) complex is phosphorylated to prevent the nuclear transfer of the NF-κB mechanism that takes place. (2) A mechanism that may activate the redox-sensitive NF-κB transcription factor by reactive oxygen species (ROS), which is produced through the stimulation of pro-inflammatory cytokines at the cellular level. Therefore, it can be speculated that our results in which the MAPK signaling pathway was not inhibited but the nuclear translocation of NF-κB was suppressed in in vitro study was because DNL regulated the nuclear translocation of NF-κB by the mechanism of (1) or (2).

Herbal medicine such as DNL contains compounds with a wider range of structural diversity. LC-MS combines the separation ability of LC with the mass spectrometry function of MS, and has been proven to be one of the most common detection methods for natural products through recent studies [56]. Therefore, we proceeded to standardize the components in DNL through LC-MS. DNL contains various components such as alkaloids, glycosides, and dibenzyl compounds [14]. Among them, gigantol, a kind of dibenzyl compound, has been used as a standard for the identification of DNL in various studies [57,58]. Additionally, according to a previous study, gigantol showed anti-inflammatory effects and based on this study, gigantol was expected to be an indicator and active ingredient of DNL. We performed LC-MS analysis of gigantol to identify DNL. As a result, the retention times of gigantol and standard gigantol in DNL were the same, and these results confirmed that DNL used in this experiment was the same herbal medicine as in previous studies.

This study has certain limitations. As described above, the phosphorylation of MAPK is caused not only by pro-inflammatory cytokines but also by cell growth factors and stress. However, in this study, MAPK phosphorylation experiments on pro-inflammatory cytokines were conducted. Furthermore, the nuclear translocation of NF-κB can occur through several mechanisms, such as MAPK-induced activation, IKK complexes, and ROS. However, in this study, the most widely known MAPK/NF-κB mechanism was investigated. Therefore, in order to clarify the anti-inflammatory mechanism of DNL, it is necessary to confirm the efficacy of DNL on the expression of MEK and MKK induced by stress or growth factors in the tissue in the future. In addition, studies on the inhibition of IKK in DNL and the ROS-induced expression of NF-κB in keratinocytes are needed.

Although this study demonstrated the effect of DNL on atopic dermatitis, it did not investigate the active component of DNL. According to previous studies, DNL has various active ingredients, among which dendrobin and moupinamide are known to inhibit inflammatory cytokines through NF-κB. However, the effect of the active ingredient in DNL on AD has not been studied yet. Therefore, if the anti-inflammatory effect and symptom suppression analysis of each component for atopic dermatitis is analyzed, it will be helpful in further research on atopic dermatitis treatment. In subsequent studies, it is necessary to verify the efficacy of DNL active ingredients for atopic dermatitis.

## 4. Materials and Methods

### 4.1. Reagents

DNCB (237329), Harris hematoxylin (HHS32-1L), Eosin Y (E-4382), toluidine blue O (t3260), Protease inhibitor cocktail (P8340), phosphatase inhibitor cocktail (P0044, P5726), PMA (16561-29-8), A23187 (C7522), dimethyl sulfoxide (DMSO, D8418), silymarin (S0292) and dexamethasone (D2915) were obtained from Sigma-Aldrich (St. Louis, MO, USA). Antibodies of CD4 (ab183685) and CD8 (ab209775) were obtained from Abcam (Cambridge, UK). Polink-2 Plus AP rabbit kit (D70-18) was purchased form GBI Labs (Bothell, WA, USA). T-PER^TM^ tissue protein extract reagent (78510) and NE-PER™ nuclear and cytoplasmic extraction reagent (78835) were purchased form Thermo Fisher Scientific (Waltham, MA, USA). Nitrocellulose blotting membrane (10600002) was obtained GE healthcare (Buckinghamshire, UK). Dulbecco’s modified Eagle’s Medium (DMEM, LM 001-05) and Dulbecco’s phosphate buffered saline (DPBS, LB 001-02) were purchased from Welgene (Welgene Bio Co., Gyeongsan, Korea). Iscove’s Modified Dulbecco’s Medium (IMDM, 31980030) and penicillin-streptomycin (P/S, 15140122) were obtained from Gibco (Grand Island, NY, USA). Fetal bovine serum (FBS, F-0500-A) was purchased from Atlas Biologicals (Fort Collins, CO, USA). TNF-α (285-IF) and IFN-γ (210-TA) were purchased from R&D system (Minneapolis, MN, USA). MTS (G5421) solution was obtained from Promega (Madison, WI, USA). RNAiso Plus (9108) was obtained from TaKaRa Bio (Tokyo, Japan). A KAPA Taq DNA polymerase kit was purchased from Kapa Biosystems (Wilmington, MA, USA). TARC, IL-6 and GAPDH primers were obtained from Genotech (Daejeon, Korea). GM-CSF (555126), MCP-1 (555179), IL-6 (555220) and TNF-α (555212) enzyme linked immunosorbent assay (ELISA) kit were purchased from BD bioscience (NJ, USA). Phosphorylation (p)-ERK (4370), ERK (4695), p-JNK (4668), JNK (9258), p-p38 (4511), p38 (9212), p-NF-κB (3033), IκBα were obtained from Cell Signaling Technology (Danvers, MA, USA). Lamin B (sc 374015) and anti β actin (sc 8432) were obtained from Santa Cruz Biotechnology (Santa Cruz, CA, USA). The ECL solution (RPN2106) was obtained from GE Healthcare Life Sciences (Seoul, Korea).

### 4.2. Preparation of Dendrobium Nobile Lindley (DNL)

DNL was purchased from Omnihub (Seoul, Korea). A total of 100 g DNL in 1 L of 30% ethanol was sonicated 3 times a week for 1 h, for 2 weeks. This solvent was filtered using gauze and filter paper, concentrated using a vacuum concentrator, and then freeze-dried to obtain 2.5 g of sample (yield: 2.5%). The extracts were stored at −20 °C until use.

### 4.3. Animals

The Balb/c mice (6 weeks old, male) were purchased from YOUNG Bio (Seongnam, Korea). All animals were approved by the Kyung Hee University Animal Care and Use Committee (KHSASP-20-184) and were raised for 26 days in the animal laboratory of Kyung Hee University. All mice were bred under standard conditions (22 ± 2 °C temperature and 50 ± 10% humidity under the 12 h light/dark cycle). The condition of all animals was checked daily, and their body weight was measured once a week. The criteria for animal euthanasia were if the animal was unconscious and did not respond to external stimuli, if its weight decreased by 20% or more, and if the animal found it difficult to consume food or water for more than 2 days due to walking discomfort. No animals died during the experiment. 

### 4.4. Sensitization and Drug Treatment in Balb/c Mice

Before the primary DNCB sensitization, the dorsal skin of mice was shaved with a clipper and then stabilized overnight. When shaving, 5% isoflurane was mixed with 100% oxygen, anesthetized using an inhalation anesthetic, and maintained with 2–2.5% isoflurane. In total, 1% DNCB was diluted in acetone/olive oil (3:1) ratio and applied to the back at 200 μL for 3 days for primary sensitization. After a latency of 4 days, 0.5% DNCB was applied to the dorsal skin at intervals of 2–3 days for secondary sensitization. DNL diluted to concentrations of 1 and 10 mg/mL in PBS/olive oil (9:1) was applied to the back skin every day. According to a previous study, natural products showed improvement in AD when 200 μL of 1~10 mg/mL concentration was applied [59,60]. Therefore, we determined the concentration of DNL by referring to the results of these previous studies. The normal group was treated with 9:1 phosphate-buffered saline (PBS)/olive oil during the 15 days. On the day when the secondary sensitization and DNL were applied together, DNL was applied 2 h after the secondary sensitization. The amount of DNCB and DNL applied to the skin was 200 μL. On the 23rd day, animals were sacrificed by continuous exposure to 5% isoflurane. A total of 0.8~1 mL of blood was collected through the cardiac puncture. Next, it was confirmed that the heart had stopped completely. After sacrifice, the liver was collected and weighted.

### 4.5. Scratching Behavior and Scoring of Atopic Dermatitis (SCORAD) Index

On the 22nd day of the animal experiment, the scratching behavior characteristics of Balb/c mice were observed before DNCB was applied. Balb/c mice were stabilized for 5 min by putting one each in a behavior cage made of a transparent acrylic plate. In the stabilized Balb/c mice, the scratching behavior was recorded by a video camera for 20 min, and the scratching behavior was counted by the observer. Mice mostly showed continuous scratching behavior. We measured all the scratching behaviors of mice using their paws. All scratching behaviors were observed in a dark room.

To measure the SCORAD index the day before sacrifice, the dorsal skin of mice was captured using a camera. To capture the dorsal skin of Balb/c mice, anesthesia was performed by inhalation with 5% isoflurane with 100% oxygen and maintained with 2–2.5% isoflurane. SCORAD is the first standardized method for the severity of atopic dermatitis in a 1993 European task force on atopic dermatitis (ETFAD) consensus report [28]. In this study, using a modified SCORAD that was partially modified from SCORAD, erythema, edema, oozing, scratches, skin thickness, and dryness of the dorsal skin of mice induced with skin lesions were evaluated through visual observation for a total of 6 items. The score was determined as no symptoms = 0, weak = 1, moderate = 2, and very severe = 3. The sum of the scores for each item was determined as the final grade, and the scores ranged from 0 to 18.

### 4.6. Cell Culture

Human keratinocyte cell-line HaCaT cells were supplied by the cell-line service (Eppelheim, Germany). HaCaT cells were cultured in DMEM supplemented with 10% FBS, 1% P/S at 37 °C, 5% CO_2_ atmosphere, and constant humidity. HMC-1 cells were supplied by Type Culture Collection (ATCC, Manassas, VA, USA). HMC-1 cells were cultured in IMDM supplemented with 10% FBS, 1% P/S at 37 °C, 5% CO_2_ atmosphere, and constant humidity. In an in vitro experiment, DNL was dissolved in DMSO at a concentration of 100 mg/mL. Although DMSO is a solvent widely used in experiments and biological fields, it is known that the ratio used for drug dissolution should be maintained at a minimum of 1% due to its toxicity [61]. Therefore, in all experiments conducted in this study, DMSO did not exceed 1% of the total medium.

### 4.7. ELISA

#### 4.7.1. In Vivo Experiment

The serum was obtained by centrifugation at 4000 rpm for 5 min after blood collection from Balb/c mice. To measure inflammatory cytokines in skin tissue, harvested skin tissue was frozen immediately after sacrifice. The frozen skin tissue was homogenized with 1 mL of T-PER^TM^ tissue protein extraction reagent (contain protease inhibitor, phosphatase inhibitor 2, and 3 cocktails). This was followed by spinning at 13,200 rpm for 20 min, and the supernatant was used for the experiment. The expression of ALT and AST, IgE in serum, and IL-6 and IL-4 in tissues was detected with an ELISA kit. The ELISA protocol followed the method provided by the kit manufacturer.

#### 4.7.2. In Vitro Experiment

HaCaT cells were stabilized for 24 h after seeding in 6-well plates 1 × 10^6^ cell/2 mL. After pretreatment with DNL at various concentrations (62.5, 125, and 250 μg/mL) or positive control (dexamethasone 10 μM and silymarin 25 μg/mL) for 1 h, TNF-α/IFN-γ (10 ng/mL each) was treated and reacted in an incubator for 24 h at 37 °C in CO_2_. HMC-1 cells were seeded in 24-well plates 3 × 10^5^ cell/500 μL and stabilized for 1 h. DNL pretreatment was applied at various concentrations (50, 100, and 200 μg/mL) or positive control (dexamethasone 10 μM and silymarin 25 μg/mL) for 1h. After pretreatment, PMA (25 nM) and A23187 (1 μM) were treated and incubated for 7 h at 37 °C in CO_2_. Inflammatory cytokines and chemokines, such as TARC, IL-6, GM-CSF, MCP-1, and TNF-α were detected with an ELISA kit. The ELISA protocol followed the method provided by the kit manufacturer.

### 4.8. Western Blot

#### 4.8.1. In Vivo Experiment

In order to evaluate MAPK protein expression in tissues, we homogenized 100 mg of frozen skin tissue with 1 mL of T-PER^TM^ tissue protein extraction reagent containing protease inhibitor and phosphatase inhibitor 2 and 3 cocktails mixture. The homogenized skin tissue was centrifuged at 13,200 rpm for 20 min and the supernatant was collected. Protein quantification was performed by bicinchoninic acid (BCA) assay, and the protein sample (35 μg) was separated by 10% sodium dodecyl sulfate-polyacrylamide gels. The separated protein was transferred to a nitrocellulose membrane, and the membrane was blocked with 5% skimmed milk. These products were incubated overnight at 4 °C with ERK, p-ERK, JNK, p-JNK, p38, and p-p38 antibody 1:1000 dilution. The next day, a peroxidase-conjugated secondary antibody dilution 1:10,000 was incubated for 1 h, and protein expression was observed using ECL solution. The expression level of each band was analyzed using ImageJ software (version 1.53a, National Institutes of Health, Bethesda, MD, USA).

#### 4.8.2. In Vitro Experiment

HaCaT cells seeded at 1 × 10^6^ cell/2 mL were stabilized for 24 h and pretreated with DNL (62.5, 125, and 250 μg/mL) for 1 h. Next, TNF-α/IFN-γ (each 10 ng/mL) was treated for 15 min (whole protein) and 5 min (nuclear and cytosol). The cells were lysed with RIPA buffer (50 mM Tris-Cl, 150 mM NaCl, 1% NP-40, 0.5% sodium deoxycholate, and 0.1% SDS) and protease inhibitor and phosphatase inhibitor 2 and 3 cocktails mixture for 30 min on ice. Nuclear and cytoplasmic were fractionated using the NE-PER Nuclear and Cytoplasmic Extraction kit. Protein quantification was performed by bicinchoninic acid (BCA) assay, and the protein sample (35 μg) was separated by 10% sodium dodecyl sulfate-polyacrylamide gels. The separated protein was transferred to a nitrocellulose membrane, and the membrane was blocked with 5% skim milk. These products were incubated overnight at 4 °C with ERK, p-ERK, JNK, p-JNK, p38, p-p38, p-NF-κB, IκB, Lamin B and Actin antibody 1:1000 dilution. The next day, peroxidase-conjugated secondary antibody dilution 1:10,000 was incubated for 1 h, and protein expression was observed using ECL solution. The expression level of each band was analyzed using ImageJ software.

### 4.9. Histological Analysis

Dorsal skin collected after the sacrifice was fixed in 10% NBF, fixed overnight, and then washed for 24 h. Dorsal skin tissue was embedded with paraffin and sectioned with a microtome (Leica Biosystems, RM2125 RTS, Wetzlar, Germany) to a thickness of 5 μm. Before observing immune cell infiltration, the sectioned skin tissue was hydrated by immersion in xylene, 100% ethanol, 90% ethanol, 70% ethanol, and deionized water. To observe infiltration of eosinophil, hydrated skin tissue was stained with Harris hematoxylin for 3 min. After washing with running water for 5 min, it was immersed in 0.5% ammonia water for 5 s and stained with eosin for 25 s. To observe infiltration of mast cell, hydrated skin tissue was stained with toluidine blue for 3 min and then washed with running water for 10 min. The stained skin tissues were dehydrated in the order of 70% ethanol, 90% ethanol, 100% ethanol and xylene, and then sealed using balsam. All stained tissues were observed through an optical microscope (BX51, Olympus, Tokyo, Japan). The thickness of the epidermis and dermis was measured in nine fields at 100× magnification and averaged. Mast cells were counted after three-field counting at 200× magnification per tissue and summed. To count eosinophils, ten fields of view were checked at 400× magnification per tissue and then summed.

### 4.10. Immunohistochemistry (IHC)

For heat-induced epitope retrieval (HIER), skin tissue sectioned at 5 μm was heated for 9 min in 0.01 M sodium citrate buffer, a specific buffer, and cooled for 30 min. After washing the tissue three times with tris-buffered saline with 0.1% tween^®^ 20 detergent (TBST), it was reacted with 0.3% H_2_O_2_ in methanol at room temperature for 30 min to inhibit peroxidase activity. Blocking was performed for 1 h with 10% normal goat serum. Next, it was washed 3 times with TBST and treated with CD4 and CD8 antibodies at a ratio of 1:1000, respectively, overnight. The expression of CD4^+^ T cells and CD8^+^ T cells was confirmed using the Polink-2 Plus AP rabbit kit, and the experimental method followed the manual provided within the kit. The tissue background was stained with Harris hematoxylin. All stained tissues were analyzed by counting and summing CD4 and CD8 in 10 fields at 400× magnification per tissue through an optical microscope (BX51, Olympus, Tokyo, Japan).

### 4.11. Cell Viability

HaCaT cell viability was evaluated by an CCK-8. HaCaT cells were seeded in 96-well plates at 1.5 × 10^4^ cell/100 μL of cell culture medium. After stabilization overnight, the HaCaT cells were exchanged with a serum-free medium containing DNL for each concentration (31.2, 62.5, 125, 250, and 500 μg/mL). After exposure to DNL for 24 h, 10 μL of CCK-8 solution was put into each well and incubated for 2 h. Next, the cell viability was measured at 450 nm absorbance by an ELISA reader (Versamax; Molecular Devices, LLC; San Jose, CA, USA). The evaluation of cytotoxicity of HMC-1 cells was measured by MTS assay. HMC-1 cells were seeded at 1 × 10^5^ cells/100 μL in 96-well plates and then stabilized overnight. After stabilization overnight, the HMC-1 cells were exchanged with a serum-free medium containing DNL for each concentration (31.2, 62.5, 125, 250, and 500 μg/mL). Subsequently, 20 μL of MTS solution was treated and incubated for 2 h, and absorbance was measured at 490 nm using an ELISA reader.

### 4.12. Reverse Transcription-Quantitate Polymerase Chain Reaction (RT-qPCR)

HaCaT cells were seeded in 6-well plates at 1 × 10^6^ cell/2 mL and stabilized for 24 h. After 1h pretreatment with DNL at concentrations of 62.5, 125, and 250 μg/mL, respectively, 10 ng/mL of TNF-α/IFN-γ was treated for 24 h in HaCaT cells. The stimulated HaCaT cells were washed three times with DPBS, and total RNA was extracted using Trizol reagent according to the manufacturer’s instructions. After extraction of RNA, 2 μg of RNA was converted to Complementary Deoxyribonucleic Acid (cDNA) using SuperScript ii reverse transcriptase (Invitrogen; Thermo Fisher Scientific, Inc.; Waltham, MA, USA), and cDNA was amplified using Taq polymerase primers. The primer sequences and reaction conditions of TARC and GAPDH are described in Table 1. These results were electrophoresed on 1.2% agarose gel with SYBR green (Invitrogen), and the band was captured using NαBI (Neoscience; Geumcheon-gu, Seoul, Korea). Target gene expression was measured by ImageJ and analyzed based on the expression of GAPDH.

### 4.13. Liquid Chromatography–Mass Spectrometry (LC–MS)

The DNL extract was analyzed by liquid chromatography–mass spectrometry (LC-MS). The separation was carried out using Xbridge C18 (250 × 4.6 mm, 5 μm). The flow rate was maintained at 0.3 mL/min, and the column temperature was 30 °C, for a total of 6 min. The mobile phase used was composed of acetonitrile with 0.1% formic acid and water with 0.1% formic acid. The elution gradient established was from 5% B over 0 min, from 5% to 50% B over 30 min, from 50% to 95% B over 60 min. MS data were acquired in the positive scan mode (mass range *m*/*z* 100–300 da).

### 4.14. Statistical Analysis

All experiments were repeated at least three times. Statistical analysis was performed using Graphpad PRISM software (v7.00, Graphpad software Inc., San Diego, CA, USA), and the experimental results were expressed as mean ± standard error (SEM). The significance of each result value was measured by one-way ANOVA method and post-tested by Dunnett’s test method. When the *p* value was less than 0.05, it was judged to be significant.

## 5. Conclusions

This study showed that DNL decreased scratching behavior and SCORAD index due to its inhibition of cytokines such as IgE, IL-6, and IL-4 in vivo. Furthermore, it was confirmed that the thickness of the epidermis and dermis was also reduced when DNL was applied by reducing the infiltration of immune cells, such as mast cells and eosinophils. In addition, this study suggested the possibility that DNL may regulate the inflammatory response in vivo through the inhibition of MAPK protein expression. In vitro, DNL inhibited the activation of MAPK and NF-κB signaling pathways. Through MAPK/NF-κB pathway inhibition, DNL inhibited the expression of TARC, GM-CSF, MCP-1, IL-6, and TNF-α in HaCaT cells. In addition, the expression of GM-CSF, TNF-α, MCP-1, and IL-6 in HMC-1 cells was also reduced by DNL. Therefore, we demonstrated that DNL can improve the symptoms of AD. Furthermore, according to the results of these studies, DNL shows potential as an alternative to treat AD while minimizing the most problematic side effects as an AD treatment agent.

## Figures and Tables

**Figure 1 ijms-23-04470-f001:**
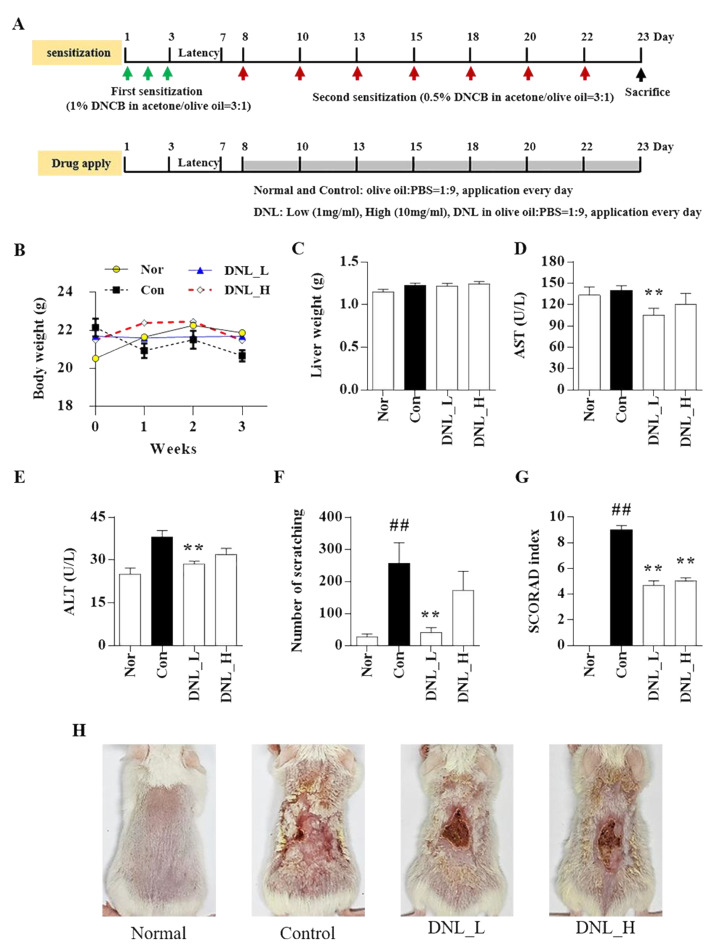
The toxicity of DNL and effect of DNL on scratching behavior and SCORAD index in DNCB-treated Balb/c mice. (**A**) Experimental schedule for topical application of DNL to dorsal skin. An amount of 1% DNCB was applied for three consecutive days to proceed with the first sensitization and had an incubation period of 4 days. After the incubation period, 0.5% DNCB was applied once every 2 or 3 days, DNL was applied every day, and sacrifices were performed on the 23rd day of the experiment. (**B**) The body weight of mice was measured using an electric scale once a week. (**C**) Liver weights were measured at the end of the animal experiment on the Balb/c mice DNL. (**D**) On the day of sacrifice, blood was collected through cardiac puncture, and serum was obtained by centrifugation. AST was confirmed by ELISA with the serum. (**E**) Serum ALT concentration was measured by ELISA. (**F**) The scratching behavior was photographed for 20 min the day before sacrifice, and the number of scratches was counted by the observer. (**G**) For the SCORAD index, six items were evaluated according to the atopic dermatitis severity standardization method, and the scores were summed. (**H**) Representative images of normal group (normal) and DNCB induction group (control), DNCB application and low-DNL-concentration-treatment group (DNL_L), DNCB application and high-DNL-concentration treatment group (DNL_H). The data are expressed as mean ± SEM (*n* = 3). ## *p* < 0.01 compared with the normal group. ** *p* < 0.01, compared with the control group.

**Figure 2 ijms-23-04470-f002:**
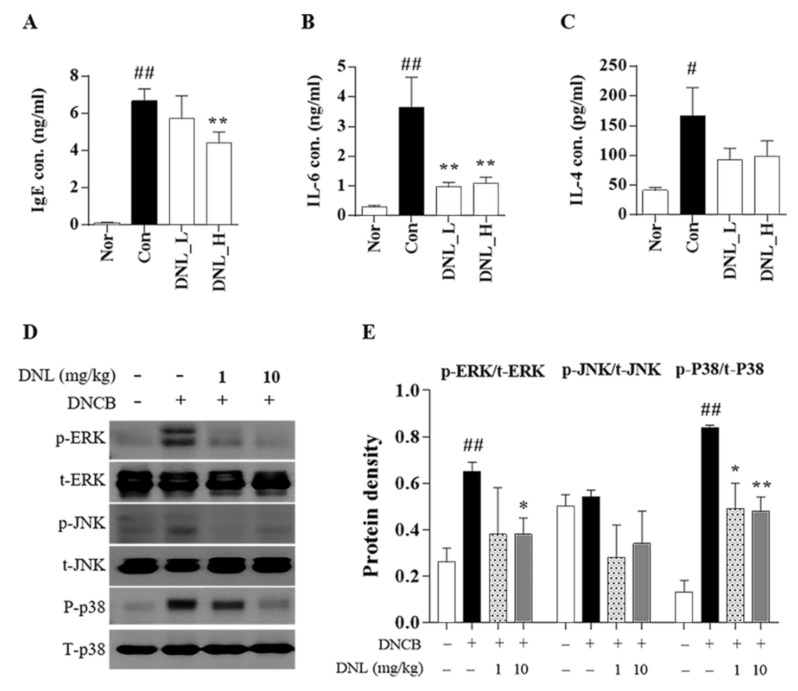
Effect of inflammatory cytokines and MAPK phosphorylation of DNL in DNCB-treated Balb/c mice. (**A**) Levels of IgE were measured by ELISA in serum. (**B**,**C**) The levels of IL-4 and IL-6 were measured by ELISA in the skin. (**D**) MAPK protein expression in DNCB-induced Balb/c mice skin was confirmed by Western blot. (**E**) MAPK was normalized to the total form of each indicator. The data are expressed as mean ± SEM (*n* = 3). # *p* < 0.05 and ## *p* < 0.01, compared with the normal group. * *p* < 0.05 and ** *p* < 0.01, compared with the control group.

**Figure 3 ijms-23-04470-f003:**
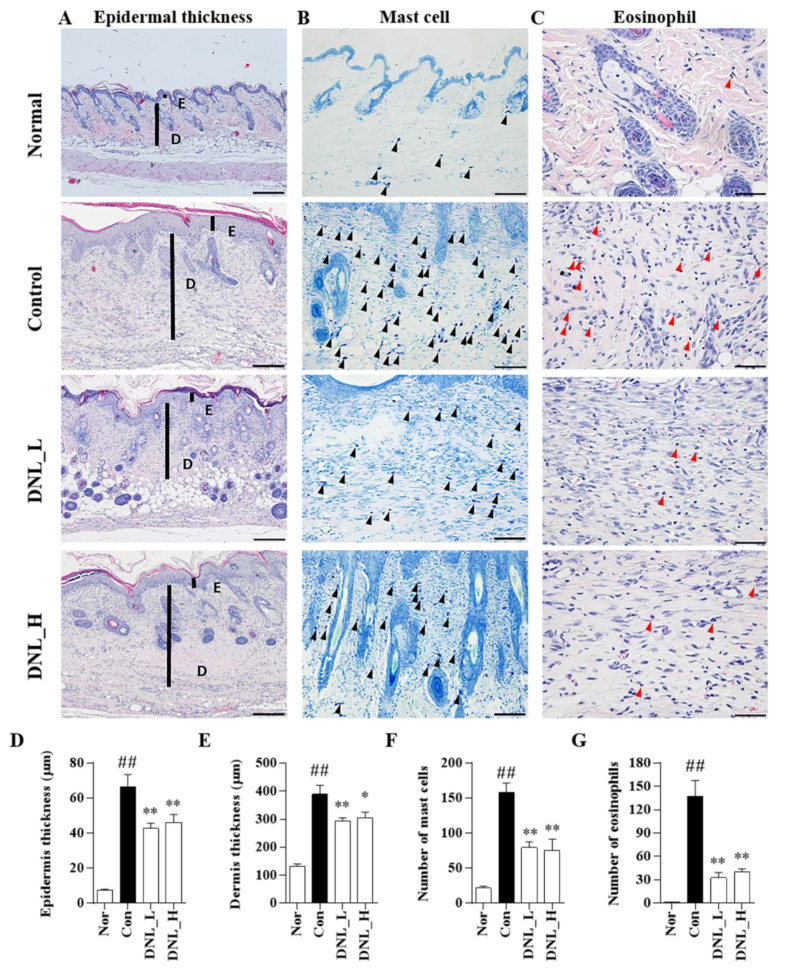
Effects of DNL on inhibition of immune cell invasion and reduction of skin thickness in DNCB-applied Balb/c mice. (**A**) To measure the thickness of the epidermis and dermis of DNCB-induced Balb/c mice, skin tissue sections were stained with H&E (magnification 100×; scale bar 200 μm). (**B**) To check the infiltration of mast cells (black arrows) in the dermis of DNCB-induced Balb/c mice, skin sections were stained with toluidine blue (magnification 200×; scale bar 100 μm). (**C**) To confirm the infiltration of eosinophils (red arrows) in the dermis of DNCB-induced Balb/c mice, skin sections were stained with H&E (magnification 400×; scale bar 50 μm). (**D**,**E**) The thickness of the epidermis and dermis was captured in three fields at a magnification of 100× per tissue. Next, three parts were measured per field of view and the values were averaged. (**F**) The number of infiltrated mast cells was counted and summed in three fields at a magnification of 200× per tissue. (**G**) The number of infiltrated eosinophils was counted in 10 fields at a magnification of 400× per tissue and summed. The data are expressed as mean ± SEM (*n* = 3). ## *p* < 0.01, compared with the normal group. * *p* < 0.05 and ** *p* < 0.01, compared with the control group. E, epidermal thickness; D, dermal thickness.

**Figure 4 ijms-23-04470-f004:**
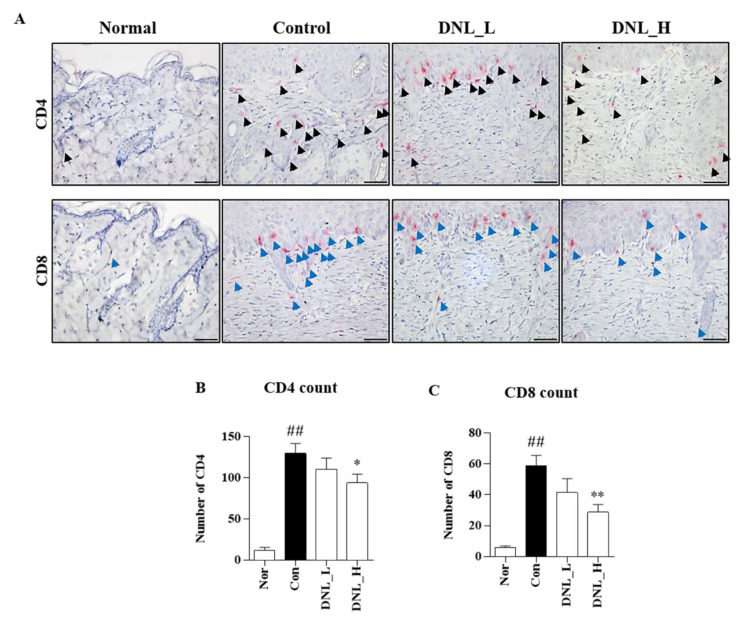
Effect of DNL in DNCB-induced Balb/c mice on CD4, CD8 T cell infiltration. (**A**) CD4 (black arrows) and CD8 (blue arrows) infiltration were confirmed in DNCB-induced Balb/c mice skin tissue by IHC (400×, scale bar 50 μm). (**B**,**C**) The numbers of CD4 and CD8 were counted and summarized in 10 fields at a magnification of 400× per tissue. The data are expressed as mean ± SEM (*n* = 3). ## *p* < 0.01, compared with the normal group. * *p* < 0.05 and ** *p* < 0.01, compared with the control group.

**Figure 5 ijms-23-04470-f005:**
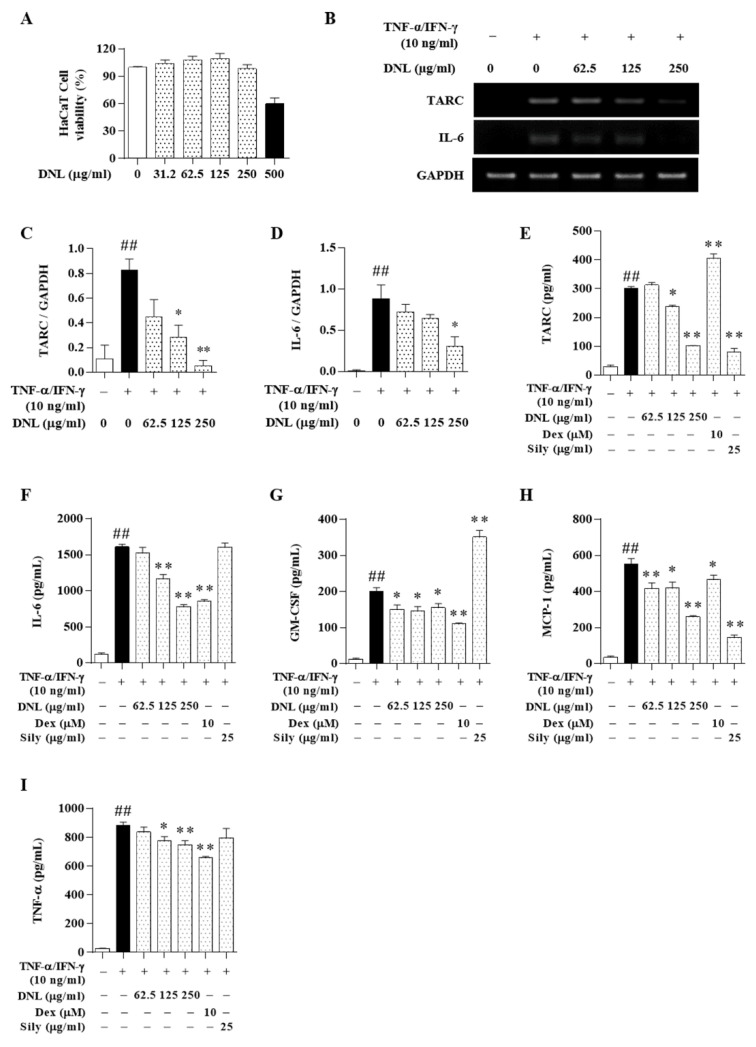
Effects of DNL on the expression of inflammatory chemokines and cytokines in TNF-α/IFN-γ-induced HaCaT cells. (**A**) HaCaT cells were treated with DNL at various concentrations for 24 h, followed by CCK-8 analysis. (**B**) TARC and IL-6 gene expression in HaCaT cells was evaluated by RT-PCR. (**C**,**D**) TARC and IL-6 were normalized to glyceraldehyde 3-phosphate dehydrogenase (GAPDH) through the ImageJ program. (**E**–**I**) The levels of inflammatory chemokines and cytokines were confirmed in HaCaT cells by ELISA. The data are expressed as mean ± SEM (*n* = 3). ## *p* < 0.01, compared with the normal group. * *p* < 0.05 and ** *p* < 0.01, compared with the control group.

**Figure 6 ijms-23-04470-f006:**
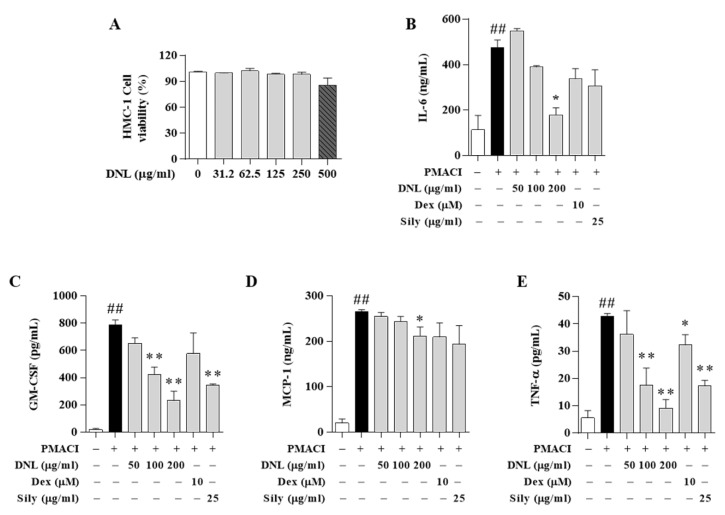
Effects of DNL on inflammatory chemokines and cytokines in PMACI-induced HMC-1 cells. (**A**) Viability of DNL in HMC-1 cells. After treating HMC-1 cells with various concentrations of DNL, they were reacted for 24 h, and then MTS analysis was performed. (**B**–**E**) Expression of inflammatory chemokines and cytokines was confirmed by ELISA in PMACI-induced HMC-1 cells. The data are expressed as mean ± SEM (*n* = 3). ## *p* < 0.01, compared with the normal group. * *p* < 0.05 and ** *p* < 0.01, compared with the control group.

**Figure 7 ijms-23-04470-f007:**
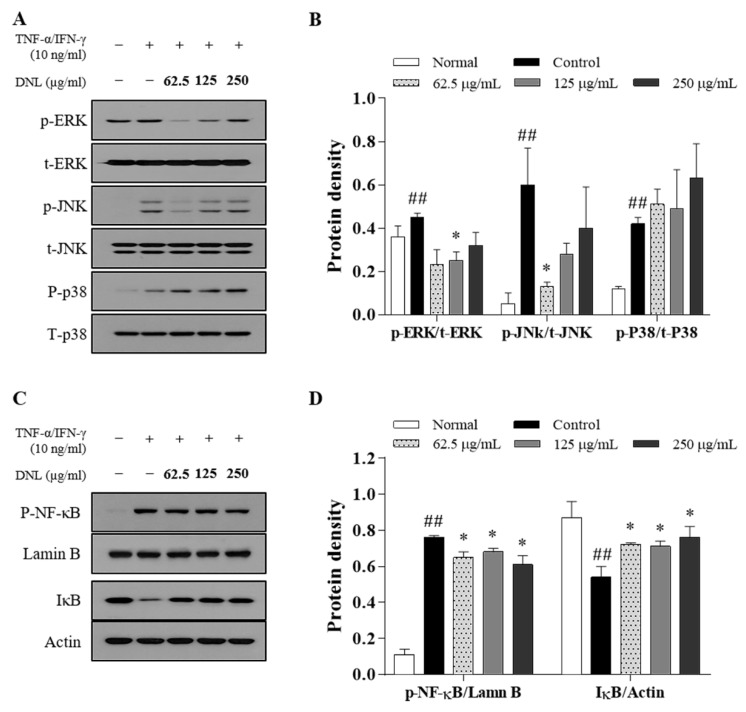
Effect of DNL on protein expression of MAPK and NF-κB/IκB in TNF-α/IFN-induced HaCaT cells. (**A**) MAPK protein expression in TNF-α/IFN-γ-induced HaCaT cells was confirmed by Western blot. (**B**) p-ERK, p-JNK, and p-p38 were normalized using the ImageJ program by T-ERK, t-JNK, and t-p38, respectively. (**C**) The phosphorylation of the signaling pathway of NF-κB/IκB in TNF-α/IFN-γ-induced HaCaT cells was confirmed by Western blot. (**D**) Phosphorylation of p-NF-κB was normalized by Lamin B, and IκB was normalized by Actin. Normalization was performed through the ImageJ program. The data are expressed as mean ± SEM (*n* = 3). ## *p* < 0.01, compared with the normal group. * *p* < 0.05 compared with the control group.

**Figure 8 ijms-23-04470-f008:**
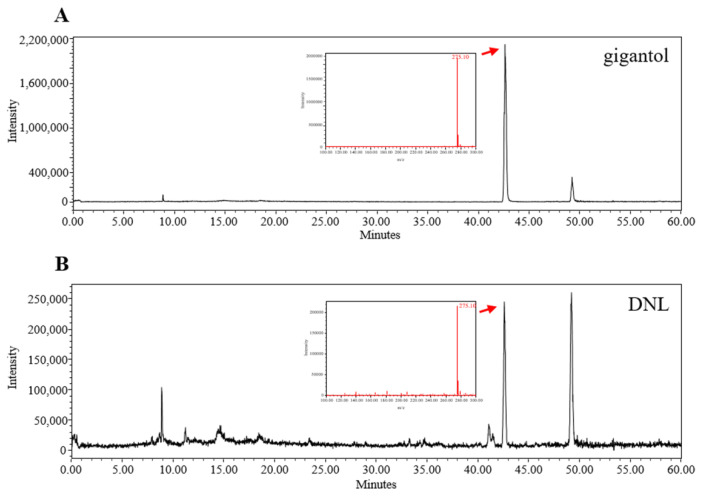
Liquid chromatography–mass spectrometry (LC–MS) of (**A**) gigantol and (**B**) DNL.

**Table 1 ijms-23-04470-t001:** Primer sequence and PCR conditions.

Gene	Primers	Sequence	Annealing Tm (°C)	Cycle	Accession Number
TARC	Forward	5′-ACT GCT CCA GGG ATG CCA TCG TTT TT-3′	57.5 °C	44	NM_002987.3
	Reverse	5′-ACA AGG GGA TGG GAT CTC CCT CAC TG-3′			
IL-6	Forward	5′-GAT GGC TGA AAA AGA TGG ATG C-3′	59 °C	45	NM_000600.4
	Reverse	5′-TGG TTG GGT CAG GGG TGG TT-3′			
GAPDH	Forward	5′-CGT CTA GAA AAA CCT GCC AA-3′	50 °C	30	NM_001256799.3
	Reverse	5′-TGA AGT CAA AGG AGA CCA CC-3′			

IL-6, interleukin 6; TARC/CCL17, thymus-and activation-regulated chemokine; GAPDH, glyceraldehyde-3phosphate dehydrogenase.

## Data Availability

Not applicable.

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
