# Peer review of "Dendrobium nobile* Lindley Administration Attenuates Atopic Dermatitis-like Lesions by Modulating Immune Cells"

_ijms, 2022, doi:10.3390/ijms23084470_

Round 1
Reviewer 1 Report
Dear authors,
Your manuscript is interesting to read.
I have some comments:
- in the abstract : the introduction on atopic dermatitis and its impact is quite extreme and the consequence on quality of life is not mentioned so far in first intention. Furthermore, did you observe a complete inhibitory/suppresive effect which is also quite strong or a reduction effect ? Conclusions should be modulated.
- introduction : same comments on AD than in abstract. Moreover, you described specific compounds in DNL but anywhere they are described as present in ethanol extract of DNL.
- results: low and hogh doses of DNL were not introduced before. Please correct ELSA for ELISA
- materials and methods (or introduction): what are the characteristics and composition of the ethanol extract of DNL ? Use of DMSO as solvent should be indicated in in vitro experiments. Was a control PBS/olive oil tested in in vivo experiments ? it is not mentioned. Why ? High and low doses of DNL extract were not precised. English languages of the sentences "TNF-alpha/IFN-gamma was treated" or "stimulated in HacaT cells shoulf be corrected.
Thanks.
Author Response
Response to Reviewer 1 Comments
We would like to thank the reviewer for your positive and very helpful comments to improve our manuscript. We addressed the comments as follows to respond point by point in the details presented by the reviewer.
in the abstract :
Point 1: The introduction on atopic dermatitis and its impact is quite extreme and the consequence on quality of life is not mentioned so far in first intention.
Response 1: We revised the effects of AD on quality of life in the abstract, and the description is as follows.
(Before the change) Atopic dermatitis (AD) is a chronic inflammatory skin disease that, in severe cases, they can lead to extreme suicidal thoughts and family breakdown. -> (After the change) Atopic dermatitis (AD) is a chronic inflammatory skin disease that can significantly affect daily life by causing sleep disturbance due to extreme itching. In addition, if the symptoms of AD are severe, it can cause mental disorders such as ADHD and suicidal ideation.
Point 2: Furthermore, did you observe a complete inhibitory/suppresive effect which is also quite strong or a reduction effect? Conclusions should be modulated.
Response 2: In this study, we observed the effects of DNL in suppressing or reducing AD-like lesions. However, in the previously written abstract, we found that the experimental results and conclusions were not clearly separated and the conclusion part was lacking.
Therefore, the revised abstract has described in detail the results of cell and animal experiments. And In the conclusion section, it was corrected that the symptoms of AD were alleviated through the anti-inflammatory effect of DNL. (page 1, Line 17-23)
introduction:
Point 3: Same comments on AD than in abstract.
Response 3: In the Introduction, we added that sleep deprivation may occur due to severe itching, one of the symptoms of AD, to further explain the effect of AD on quality of life. In order to connect the flow of content, the order of the Introduction sentences was changed.
Added of the sentence is as follows. “Also, AD can cause fatigue due to reduced sleep time and increased sleep disturbance due to extreme itching, which affects daily life”
Point 4: Moreover, you described specific compounds in DNL but anywhere they are described as present in ethanol extract of DNL.
Response 4: We acknowledge the lack of DNL explanation and results in the previous manuscript. Thanks for broadening a new perspective.
It was judged that the proof of the specific compound for the ethanol extract of DNL was insufficient, and gigantol was added to the introduction that the anti-inflammatory effect was proven among the components of DNL. (page2, Line 67-69)
In addition, in order to connect the flow of the contents, it was additionally described that the anti-inflammatory effect and the improvement of AD symptoms were closely related (page 2, Line 74-75).
We judged that the proof of specific compounds for the ethanol extract of DNL was insufficient, and we performed LC-MS using gigantol, which has been proven to have anti-inflammatory effects among the components of DNL.
We performed LC-MS using gigantol. As a result of the experiment, DNL and gigantol recorded peaks at the same retention time. This allowed us to identify DNL. These results are schematically shown in Figure 8. (page 11, Line 282)
results:
Point 5: Low and hogh doses of DNL were not introduced before.
Response 5: In this study, we used 1mg/ml of DNL_Low and 10mg/ml of DNL_H. However, we have confirmed that these are missing. Therefore, we added this information in Result_2.1 and Figure 1A. (page 2, Line 89)
Point 6: Please correct ELSA for ELISA.
Response 6: We modified ELSA for ELISA in figure 1 Legend and result 2.2. (page 3, Line 109,110 & page 4, Line 122)
materials and methods (or introduction):
Point 7: What are the characteristics and composition of the ethanol extract of DNL?
Response 7: According to the research results of “Anti-aging properties of Dendrobium nobile Lindl.: From molecular mechanisms to potential treatments”, DNL components are classified into compounds such as alkaloid glycosides and phenanthrenes, and there are about 80 types.
These contents are described in result_2.8. (Page 10, Line 273-280)
We performed LC-MS with the supporting data, and this was mentioned in the response of point 4.
Point 8: Use of DMSO as solvent should be indicated in in vitro experiments.
Response 8: In the initial manuscript, we described the use of DMSO in vitro at the end of the method_4.2 Preparation of Dendrobium Nobile Lindley (DNL) part.
To rewrite this content in detail, we classified method_4.10_Cell culture and assessment of cell viability in the previous manuscript into 4.6_cell culture and 4.11_viability respectively. And described in detail how DNL was dissolved in DMSO in the 4.6_cell culture part.
We found that the details of DMSO products were omitted from the previous manuscript, so we added them. (page 14, Line 466)
Point 9: Was a control PBS/olive oil tested in in vivo experiments? It is not mentioned. Why?
Response 9: When applying the DNL, PBS/olive oil was treated in the normal group, but this content seems to be omitted from the manuscript. Therefore, we added this part to Figure 1A and Materials and methods_4.4 Sensitization and drug treatment in Balb/c mice.
When we induced AD with DNCB, the normal group was applied 1X PBS, and control, DNL_L and DNL_H groups were applied by dissolving DNCB in acetone/olive oil.
When AD was induced with DNCB and the drug was applied, the normal and control groups were applied with PBS/olive oil, and DNL was treated by dissolving 1 or 10 mg/ml in PBS/olive oil.
But this content seems to be omitted from the manuscript. Therefore, we added this part to Figure 1A and Materials and methods_4.4 Sensitization and drug treatment in Balb/c mice. (page 3, Line 100 & page 16, Line 514-517)
Point 10: High and low doses of DNL extract were not precise.
Response 10: We found that the concentration of DNL was not accurately stated. The concentration of DNL was described in result_2.1 and figure 1A as the same answer as in comment 5. (page 2, Line 89 & page 3, Line 100)
In addition, references were added by investigating the study related to the determination of the DNL concentration.
Based on the previous study, AD was improved when the natural product was applied at a concentration of 1~10 mg/ml. Therefore we determined the concentration of DNL to be 1mg/ml and 10 mg/ml.
The above information was added to Materials and methods_4.4 Sensitization and drug treatment in Balb/c mice. (page 16, Line 514-517)
Reference
* Immunomodulatory effects of Pseudostellaria heterophylla (Miquel) Pax on regulation of Th1/Th2 levels in mice with atopic dermatitis; MOLECULAR MEDICINE REPORTS; 2017
* Immunomodulatory Activities of the Benzoxathiole Derivative BOT-4-One Ameliorate Pathogenic Skin Inflammation in Mice; Journal of Investigative Dermatology; 2016
Point 11: English languages of the sentences "TNF-alpha/IFN-gamma was treated" or "stimulated in HacaT cells shoulf be corrected.
Response 11: As followed by the reviewer's comments, we revised the sentences to "TNF-alpha/IFN-gamma was treated" or "stimulated in HacaT cells". The amended part is as follows.
Result_2.7_(Before the change) Effect of DNL on TNF-α/IFN-γ-stimulated HaCaT cell activation of NF-κB and MAPKs -> (After the change) Effect of DNL on TNF-α/IFN-γ-treated HaCaT cell activation of NF-κB and MAPKs. (page 9, Line 249)
Materials and methods_4.12_RT-PCR_ (Before the change) TNF-α/IFN-γ was stimulated for 24 h in HaCaT cells. -> (After the change) TNF-α/IFN-γ treated for 24 h in HaCaT cells. (page 18, Line 650)

Reviewer 2 Report
Atopic dermatitis is a chronic inflammatory skin disease that they can lead to extreme suicidal thoughts and family breakdown.
In the presented work, the authors evaluated the inhibitory effect of DNL on AD in mouse model and HaCaT cells. The authors provided evidences that shows DNL suppressed the immune cells such as mast cells and eosinophils, and finally alleviating the symptoms of AD.
This is a good manuscript and should be interesting to the readers of the International Journal of Molecular Sciences. The manuscript is almost ready for publication, just a suggestion.
The reviewer noted this manuscript has too many references (63). For a research article, it seems it is not good to cite too many papers.
Author Response
Response to Reviewer 2 Comments
We would like to thank the reviewer for your positive and very helpful comments to improve our manuscript. We addressed the comments as follows to respond point by point in the details presented by the reviewer.
Atopic dermatitis is a chronic inflammatory skin disease that they can lead to extreme suicidal thoughts and family breakdown.
In the presented work, the authors evaluated the inhibitory effect of DNL on AD in mouse model and HaCaT cells.
The authors provided evidences that shows DNL suppressed the immune cells such as mast cells and eosinophils, and finally alleviating the symptoms of AD.
This is a good manuscript and should be interesting to the readers of the International Journal of Molecular Sciences.
The manuscript is almost ready for publication, just a suggestion.
Point 1: The reviewer noted this manuscript has too many references (63). For a research article, it seems it is not good to cite too many papers.
Response 1: According to the reviewer's opinion, references with similar contents were abbreviated, and some references were added due to additional LC-MS experiments. Therefore, it was finally reduced from 63 references previously to 58 references.
